# Combined Treatments of High Hydrostatic Pressure and CO_2_ in Coho Salmon (*Oncorhynchus kisutch*): Effects on Enzyme Inactivation, Physicochemical Properties, and Microbial Shelf Life

**DOI:** 10.3390/foods9030273

**Published:** 2020-03-03

**Authors:** Mario Perez-Won, Roberto Lemus-Mondaca, Carolina Herrera-Lavados, Juan E. Reyes, Teresa Roco, Anais Palma-Acevedo, Gipsy Tabilo-Munizaga, Santiago P. Aubourg

**Affiliations:** 1Department of Food Engineering, Universidad del Bío-Bío, Av. Andrés Bello 720, Chillán 3780000, Chile; cpherrera@ubiobio.cl (C.H.-L.); jreyes@ubiobio.cl (J.E.R.); apa.anis@gmail.com (A.P.-A.); gtabilo@ubiobio.cl (G.T.-M.); 2Department of Food Science and Chemical Technology, Universidad de Chile, Santos Dumont 964, Independencia, Santiago 8380000, Chile; rlemus@uchile.cl; 3Department of Food Engineering, Universidad de La Serena, Av. Raúl Bitrán 1305, La Serena 1700000, Chile; trocob@userena.cl; 4Department of Food Technology, Instituto de Investigaciones Marinas (CSIC), 36208 Vigo, Spain; saubourg@iim.csic.es

**Keywords:** coho salmon, HHP + CO_2_, endogenous enzymes, lipid oxidation, refrigeration, shelf life

## Abstract

This study focused on applying different high hydrostatic pressure + carbon dioxide (HHP + CO_2_) processing conditions on refrigerated (4 °C, 25 days) farmed coho salmon (*Oncorhynchus kisutch*) to inactivate endogenous enzymes (protease, lipase, collagenase), physicochemical properties (texture, color, lipid oxidation), and microbial shelf life. Salmon fillets were subjected to combined HHP (150 MPa/5 min) and CO_2_ (50%, 70%, 100%). Protease and lipase inactivation was achieved with combined HHP + CO_2_ treatments in which lipase activity remained low as opposed to protease activity during storage. Collagenase activity decreased approximately 90% during storage when applying HHP + CO_2_. Combined treatments limited the increase in spoilage indicators, such as total volatile amines and trimethylamine. The 150 MPa + 100% CO_2_ treatment was the most effective at maintaining hardness after 10 days of storage. Combined treatments limited HHP-induced color change and reduced the extent of changes caused by storage compared with the untreated sample. Microbial shelf life was extended by the CO_2_ content and not by the HHP treatments; this result was related to an increased lag phase and decreased growth rate. It can be concluded that combining HHP and CO_2_ could be an effective method of inactivating endogenous enzymes and extend salmon shelf life.

## 1. Introduction

As global demand for marine products increases, the fishing industry is paying close attention to the development of aquaculture to satisfy consumer requirements. Among farmed fish, coho salmon (*Oncorhynchus kisutch*), also known as silver salmon, has attracted great interest due to its increasing production in countries such as Chile, Japan, and Canada [1]. Salmon is a high-value product that is very popular worldwide; however, one of the principal factors limiting the export of refrigerated seafood is its short shelf life [2].

Fish and shellfish are known to rapidly deteriorate due to their high water activity, neutral pH, high content of free amino acids and unsaturated fatty acids, and presence of autolytic enzymes. Deterioration is primarily caused by bacterial action. Loss of quality greatly depends on temperature; thus, refrigerated and frozen storage is used to inhibit physicochemical and microbial deterioration. Chemical changes in muscle tissue, such as autoxidation or enzymatic hydrolysis of proteins or fatty acids, can cause off-flavors and discoloration; in other cases, tissue proteolytic activity can generate unacceptable softening of seafood [3]. Texture is an important quality of fish fillets; it strongly affects the organoleptic quality and processability of fish muscle [4]. Fish texture is influenced by several factors, such as pH, rigor mortis, proteolysis, and storage temperature during post-mortem storage. The proteolytic cleavage of structural proteins in myofibrils and the extracellular matrix causes softening of the fish muscle during post-mortem refrigerated storage. The degradation of both collagen and myofibrillar proteins has been strongly related to tissue softening and quality deterioration during refrigerated storage [5,6]. Endogenous proteases such as calpains, metalloproteinases, lysosomal cathepsins, elastases, and collagenases significantly contribute to the detrimental post-mortem softening of fish muscle [7]. Cathepsins have reportedly caused the greatest degradation of the myosin heavy chain in several fish species. Calpains are less responsible for myofibrillar cleavage and are known to increase the susceptibility of myofibrillar proteins to other proteases [8]. Specifically, collagen acts as the scaffold for fish muscle, plays an important role in the firmness and structure of the flesh, and is responsible for muscle stability. Its disruption is mainly caused by collagenases, which are enzymes that degrade the collagen triple helix [6]. Previous studies have shown that collagenases are important for the texture of fish muscle during ice storage [9]. Therefore, reduced collagenase activity and subsequent collagen stabilization could control texture deterioration of refrigerated products by controlling the autolytic degradation of fish muscle.

The increased demand for high-quality fresh marine products has led to the development of advanced processing systems to ensure prolonged quality. These alternatives include previous chemical and physical treatments [10] and preservative packaging [11,12]. High hydrostatic pressure (HHP) processing is a promising alternative to the thermal treatment, which has proven to increase the shelf life of marine products while inactivating microbial growth and endogenous enzyme deterioration [13]. As for HHP, pressure from 100 to 1000 MPa is required to inactivate bacteria [14]; however, in the presence of CO_2_, it is possible to lower the pressure requirement to between 5 and 35 MPa [15]. Combining both technologies has been effective in deactivating microorganisms and enzymes at pressures < 50 MPa. At this pressure level, pressure alone has no effect on enzyme inactivation [16]. The overall effect depends on several factors such as pressure level, temperature, pressure time, pH, CO_2_ content, food ingredients, and storage time. Most of the work associated with the effect of HHP + CO_2_ on enzymatic activity has been conducted on vegetable and fruit juices. To our knowledge, no work has been done on the effect of HHP and CO_2_ on the enzymatic activity of fish. Given that enzymatic activity plays a crucial role in fish quality, the aim of this work was to determine the combined effect of HHP and CO_2_ on collagenase, protease, and lipase enzymatic activity, physicochemical properties, and microbial shelf life of coho salmon (*Oncorhynchus kisutch*).

## 2. Materials and Methods

### 2.1. Raw Material

The coho salmon was farmed at Salmones Aysen SA, Puerto Montt, Chile; samples were beheaded, gutted, filleted, and transported in boxes with ice to the food processing laboratory (Chillán, Chile). Each fillet weighed between 150 and 180 g; fillets were packaged and vacuum-sealed in high-density polyethylene bags. Once packaged, the samples were immediately pressurized in a process that lasted < 4 hours and they were stored at 4 °C until further analysis. To evaluate the effect of storage on the physicochemical properties of the flesh, batches were stored at 4.0 ± 1.0 °C and samples were taken on days 0, 3, 7, and 10 for analysis. For microbiological analyses, counts were performed by the combined application of high hydrostatic pressure and carbon dioxide (HHP + CO_2_) before day 0 and after 1, 3, 5, 7, 10, 12, 15, 18, 21, and 25 days of refrigerated storage at 4 °C. Samples for each treatment were in triplicate.

### 2.2. High Hydrostatic Pressure and Carbon Dioxide (HHP + CO_2_)

Selection criteria were freshness, color, size, and the absence of any mechanical damage. Fillet thickness was approximately 10.5 ± 0.2 mm. The air inside the bags of CO_2_ samples was removed and flushed with food grade CO_2_ at of 50%, 70%, and 100% CO_2_ concentrations (Indura, Chile) for 20 seconds to ensure the desired atmosphere. Bags were immediately sealed without gas loss and placed on ice until HHP processing. The HHP treatment was performed in an isostatic pressing system (Avure Inc., Kent, WA, USA) with a cylindrical pressure chamber (length: 700 mm; diameter: 60 mm). A 150 MPa pressure level with a 5-minute holding time at room temperature was used according to previous results. Eight treatments were generated according to the experimental design: 0 MPa/0% CO_2_ (0/0); 0 MPa/50% CO_2_ (0/50); 0 MPa/70% CO_2_ (0/70); 0 MPa/100% CO_2_ (0/100); 150 MPa/0% CO_2_ (150/0); 150 MPa/50% CO_2_ (150/50); 150 MPa/70% CO_2_ (150/70); 150 MPa/100% CO_2_ (150/100). Samples were placed in polyethylene bags, heat-sealed, and exposed to high-pressure treatment. Pressurized samples were immediately stored at 4 °C until further analysis.

### 2.3. Enzymatic Performance

#### 2.3.1. Protease and Lipase Activity

Protease and lipase activity at pH 7.0 was determined using the method described by Bernal et al. (2018) [17] with modifications. Enzyme extraction was carried out by mixing a salmon sample with extraction buffer consisting of sodium phosphate 50 mM and KCl 10 mM at a 1:4 ratio; ethanol at 20 g/100 g was included in the buffer for lipase extraction. The mixture was homogenized using an UltraTurrax Homogenizer (IKA T18 Digital, Staufen, Germany) at 6000 rpm for 30 to 40 seconds. Homogenates were centrifuged at 4500 rpm for 30 minutes at 4 °C and supernatant was filtered with Whatman N°1 paper. The complete process was performed using ice flakes as a coolant. Enzymatic kinetics were measured by the increase in absorbance produced by the release of p-nitrophenyl (pNP) in the hydrolysis of 0.4 mM pNPB for lipase (p-Nitrophenyl butyrate, Sigma-Aldrich) and 2.5 mM BOC-Ala-p-NP (N-(tert-butoxycarbonyl)-L-alanine, Sigma-Aldrich) for protease. To start the reaction, 50 µL enzyme extract and 20-25 µL substrate (20 µL for protease, 25 µL for lipase) were added in each cell containing 2 mL of 25 mM sodium phosphate buffer at pH 7.0. Pure buffer was used as blank. Measurements were performed at 348 nm at 30 °C for 60 seconds with a Jasco VR-730 spectrophotometer (Tokyo, Japan). Results were expressed in international units of activity (IU), defined as the amount of enzyme that hydrolyzes 1 µmol of pNPB or BOC-Ala-p-NP (lipase or protease, respectively) per minute under the above-mentioned conditions.

#### 2.3.2. Collagenase Activity

Collagenase was extracted from salmon samples using the method described by Apse et al. with modifications [18]. Salmon was homogenized with 100 mM Tris-HCl buffer (with CaCl_2_ 20 mM) at a 1:4 salmon:buffer ratio with a homogenizer (IKA, Ultraturrax T18, Staufen, Germany) for 30 seconds, checking that the temperature did not exceed 10 °C by using ice flakes. Homogenates were centrifuged at 6000 rpm for 40 minutes at 4 °C and supernatant was filtered with Whatman N°1 paper. The extracted enzyme was incubated with Tricine buffer (50 mM Tricine, 10 mM calcium chloride, and 400 mM sodium chloride, pH 7.5) for 96 hours prior to the assay. Enzymatic activity was measured with FALGPA (2-furanacryloyl-L-leucylglycyl-L-prolyl-L-alanine, Sigma-Aldrich) according to the method described by Van Wart and Steinbrink [19] with modifications. In each cell containing 1 mL of incubated enzyme, 10 uL of 0.05 mM substrate were added to start the reaction. Pure buffer was used as a blank. Measurements were taken at 345 nm and at 30 °C for 300 seconds with a Jasco V-730 spectrophotometer (Tokyo, Japan).

### 2.4. Physicochemical Properties

#### 2.4.1. Total Volatile Basic Nitrogen (TVB-N), Trimethylamine (TMA), and Thiobarbituric Acid (TBA) Values

The TVB-N content was determined in a 10 g salmon sample by steam distillation with MgO and water (50 mL) with a Kjeldahl distillation apparatus and titration with 10 mM KCl according to Gallardo et al. [20].

The TMA value was measured according to Association of Official Analytical Chemists (AOAC) official method 971.14 [21]. Homogenized salmon samples (6.25 g) were blended with 12.5 mL 7.5% trichloroacetic acid (TCA) solution, centrifuged (CENTAUR 2 MSE) at 3000 rpm, and filtered. Four milliliters of extract were placed in a test tube containing 1 mL 20% formaldehyde, followed by 10 mL anhydrous toluene and 3 mL saturated potassium carbonate solution. The test tubes were shaken vigorously and the toluene phase was transferred to a tube containing 0.2 g anhydrous sodium sulfate and shaken to eliminate water residues. A 5 mL aliquot of the water-free toluene extract was transferred to another test tube and mixed with 5 mL 0.02% picric acid solution. Absorbance of the standards and samples was measured at 410 nm with a spectrophotometer (Spectronic® 20 Genesys®, Spectronic Instruments, Chicago, IL, USA). The TMA concentration was calculated from a standard curve using TMA as substrate and was expressed as mg TMA/100 g sample.

The TBA index was determined according to Sorensen and Jorgensen [22]. Five grams of salmon sample were homogenized with 15 mL 7.5% TCA solution containing 0.1% propyl gallate and 0.1% ethylenediaminetetraacetic acid disodium salt (EDTA). Samples were filtered and 3 mL were placed in a test tube containing 3 mL 0.02 M TBA solution. The mixture was agitated and incubated at 100 °C for 40 minutes and the reaction was stopped by cooling in ice for 2 minutes. Absorbance was measured at 532 nm against a blank containing 3 mL distilled water and 3 mL TBA solution. Results were expressed as mg malondialdehyde/kg fish muscle. All measurements were done in triplicate, and analytical grade solvents and reagents were purchased from Sigma-Aldrich Company Ltd. (St. Louis, MO, USA).

#### 2.4.2. Textural Properties and Surface Color

For the texture profile analysis (TPA), salmon fillets were manually cut into cubes with a 20.2 ± 1.0 mm thickness. Ten cubes were analyzed per treatment with a texture analyzer (Stable Micro Systems TA.XTplus, Surrey, UK). Outlier measurements were removed and the results from the remaining samples (usually *n* = 10) were averaged. For TPA, the cube core was compressed to 50% with a 10-second gap between the two compressions and a test speed of 2 mm/s using a cylindrical aluminum flat probe (diameter: 25 mm). Texture analysis was automatically performed by the texture expert software (v.2.63 Stable Micro Systems Ltd.), and the hardness (N), cohesiveness (dimensionless), and springiness (cm) parameters were recorded as defined by Briones-Labarca et al. [23]. Sample surface color was measured with a Minolta Colorimeter (CM-1000, Tokyo, Japan) that was previously calibrated with white and black glass standards. Color changes were measured by colorimetric evaluation with the following parameters: lightness (L*), red/green (a*), and yellow/blue (b*). The CIE L*, a*, and b* color coordinates (considering standard illuminant D_65_ and observer 10°) were determined. The experiments were performed in triplicate. The colorimeter yielded L*, a*, and b* values for each spot, which were converted to a total color difference value (∆E) from ∆E = (∆L*^2+^∆a*^2+^∆b*^2^)^0.5^. All measurements were performed at room temperature (20 ± 1 °C).

### 2.5. Microbial Analysis and Shelf-Life Estimation

All samples were analyzed for the number of mesophilic and psychrophilic aerobic microorganisms, *Pseudomonas* spp., and lactic acid bacteria (LAB). From each sample, 20 grams was aseptically obtained and homogenized with 180 mL chilled maximum recovery diluent (Oxoid, Basingstoke, England) in a filter stomacher bag using a Stomacher 400 Circulator (Seward Laboratory, London, UK) at 230 rpm for 2 minutes. Further decimal dilutions were prepared with the same diluent and analyzed for aerobic mesophilic and psychrophilic microorganisms [24], LAB [25], *Shewanella putrefaciens* [24], and *Pseudomonas* spp. [26]. The presence of *Clostridium perfringens* was tested at the end of the storage period [27].

Microbial data were transformed into logarithms of the number of colony-forming units (log CFU/g). The detection limit was 10 CFU/g (1.0 log CFU/g), except for Pseudomonas, which was 100 CFU/g (2.0 log CFU/g). When no colonies were detected, an arbitrary value of 0.5 log CFU/g was assigned, except for *Pseudomonas* spp., which was allocated a value of 1.0 log CFU/g.

Microbiological shelf life was determined according to the description by Reyes et al. [24] in which growth curves of experimental data were fitted to the reparametrized version of the modified Gompertz equation to estimate growth kinetic parameters, including shelf life [28]. To estimate shelf life, a 6.0 log CFU/g maximum limit of acceptability for mesophilic and psychrophilic microorganisms was considered. This value is commonly used for fish species because it correlates with the onset of unpleasant odor and taste [29,30,31].

### 2.6. Statistical Analysis of Quality Parameters

The statistical analysis of experimental data was determined with the Statgraphics Plus^®^ v.5.1 software (Statgraphics Corp. 1991). An analysis of variance (ANOVA) was applied to estimate any statistically significant differences at a 95% confidence level (*p* < 0.05) together with a multiple range test (MRT) to compare data.

## 3. Results and Discussion

### 3.1. Enzymatic Activities

The enzymatic activities of coho salmon fillets are shown in Table 1. It is worth mentioning that, according to the properties of CO_2_, this allows a substantial inactivation of enzymes at relatively mild operating conditions in which the thermal treatment is not effective [32]. In addition, the CO_2_ gas is also non-toxic, non-flammable, inexpensive, and is easily removed simply by depressurization and outgassing [33].

The protease activity of the control sample was 0.397; immediately after the treatments (non-assisted CO_2_ and HHP-assisted CO_2_), this activity was reduced by approximately 30%, 50%, and 60% for the 0/50, 0/70, and 150/100 treatments, respectively. Despite this reduced activity on day 0, protease activity values did not show a clear pattern during storage from day 3 to day 10. Therefore, protease activity on day 10 was considered as the final storage result in which treatments 0/50, 0/100, and 150/70 could maintain reduced protease activity by approximately 50% to day 10. From a statistical point of view, ANOVA results comparing protease activity values at a 95% confidence level showed a significant influence of the treatments on this parameter (*p* < 0.05). An MRT was then performed to determine the significant means among treatments on day 0, and showed significant differences between the protease activity values; however, a single homogeneous group was obtained (i.e., 0/0-50/0-150/70), which demonstrates that pressure at 150 MPa could not inhibit enzyme activity. However, there was no clear trend at the end of storage for protease activity and four homogeneous groups were found: 0/0-0/70-150/0-150/70, 0/50-0/100, 150/50, and 150/100. Despite the initial decrease in protease activity during storage, the original protease activity was recovered in most cases and even increased after 10 days. This could indicate that structural changes in proteases from coho salmon produced by HHP and CO_2_ are reversible under these conditions. Lakshmanan et al. [34] found an initial modification in the proteolytic activity of cathepsins B and L after the HHP treatment and a recovery of enzymatic activity during storage. It has been proposed that pressures up to 150 MPa can induce changes in protein quaternary structure and pressures > 300 MPa are required to modify the secondary structure and cause protein denaturation [13]. According to these results, the combination of HHP at 150 MPa and CO_2_ up to 100% was not convenient to completely inactivate the enzymes; therefore, higher pressures would be required to irreversibly deactivate salmon proteases. The effect of CO_2_ in enzyme inactivation is believed to be a pH effect, but no correlation between enzyme activity and pH could be implied in the present study (Table 2).

Lipases are enzymes that degrade phospholipids into free fatty acids and glycerol. The increase in fatty acids has been related to quality deterioration due to the promotion of lipid oxidation and protein denaturation [35]. Applying the HHP + CO_2_ combination increased lipase activity immediately after treatment, but decreased during storage. In addition, the statistical analysis showed significant differences between the lipase activity values for each storage day at the 95% confidence level. When evaluating the effect of the storage day on lipase activity (maintaining the treatment constant), only 0/50 and 150/100 showed *p* > 0.05, that is, there was no statistically significant difference for lipase activity for the duration of storage (95% confidence level). On the basis that the fatty acids present in the salmon samples vary between C16 and C22 in chain length, this phenomenon can be due to the fact that lipases hydrolyze the shorter chain length fatty acids first and then those of the longer chains. Thus, given the activity of the control sample, it is possible to infer that activity decreases on day 10 because the shorter fatty acid chain has already been degraded by salmon lipases. Teixeira et al. [36] found that by applying 100 MPa at high pressurization rates (14 MPa/s), lipase activity increased in sea bass fillets, but this increased lipid degradation was not associated with the loss of quality parameters.

No collagenase activity was observed on day 0 in any sample. Enzymatic activity increased during storage to 2.458 in the control sample, but when applying 150 MPa, the collagenase activity decreased by up to 90%. In addition, based on the ANOVA performed to evaluate collagenase activity values at a 95% confidence level, statistically significant differences were implied. The combined treatments were more effective in reducing this activity than using CO_2_ alone (Table 1). The ANOVA showed a p-value < 0.05, indicating a statistically significant difference between the collagenase activity values (day 10/days) against different treatments (α = 95%). An MRT test was developed, which generated five homogeneous groups (0/0, 0/50, 0/100-150/0-150/100, 0/70-150/50, and 150/50-150/70); the 0/100-150/0-150/100 group had a greater decrease in collagenase activity. The lowest collagenase activity at the end of storage was achieved with the HHP and 100% CO_2_ combination.

The sorption of dense CO_2_ by the enzyme molecule would usually cause conformational changes resulting in loss of activity. Specifically, decreased activity could be related to the modification of secondary structures in enzyme molecules [37]. Slightly low pH values (5.9–6.4) were found to benefit enzymatic inactivation. Most of the work regarding HHP + CO_2_ has been performed on vegetable and fruit juices. Duong and Balaban [38] showed that pressure time had a significant effect on pectin methylesterase (PME), polyphenol oxidase (PPO), and peroxidase (POD) enzyme activities in feijoa puree, whereas pressure only had a significant effect on PPO inactivation.

### 3.2. Chemical Indices

Chemical changes associated with quality loss in coho salmon fillets were measured by TVB-N, TMA, and TBA assays (Figure 1). The TVB-N is often used as a spoilage indicator and includes the total sum of ammonia, dimethylamine, TMA, and other basic volatile nitrogenous compounds [39]. All values were lower than those indicated by technical standards (30 mg N/100 mg fish muscle). The TVB-N values immediately increased after CO_2_ treatments, except for the 0/50 treatment in which no significant difference was observed compared with the control (Figure 1). The HHP treatment alone had no effect on TVB-N values; however, when combined with CO_2_, TVB-N content increased. A marked increase of TVB-N content was detected when no CO_2_ was applied; on the contrary, treatments with 100% CO_2_, with and without pressure, inhibited this increase.

The TMA content is often used as a quality and shelf-life indicator of fish and seafood because it is related to fish odor [23]. All samples showed low TMA values that were under the acceptable limit of 5 mg/100 g, which is considered adequate for different fish species [24]. The HHP and CO_2_ treatments had no significant effect on TMA values on day 0. The TMA increased during storage, especially when no pressure was applied. Treatments with pressure and 0%, 70%, and 100% CO_2_ were effective in controlling the increase in TMA during the first 7 days of storage. The most effective treatment was the one that used 150 MPa without CO_2_ and in which TMA decreased after 10 days. Previous studies have shown that HHP is effective in inhibiting TMA production in different fish species [24,29,31]; this is attributed to the inhibitory activity of HHP on TMA-producing bacteria such as *S. putrefaciens* and *Photobacterium phosphoreum* even at pressures <100 MPa [40].

Lipid oxidation in coho salmon fillets was measured by the thiobarbituric acid reactive substance (TBARS) assay. The CO_2_ treatment increased the TBA value only when the highest concentration was applied, and HHP had a significant impact on the TBA index in treatments without CO_2_ and with 50% CO_2_. This indicates that the HHP treatment alone increased lipid oxidation; the degree of oxidation decreased when a high CO_2_ concentration was included. At the end of the evaluation period, samples subjected to HHP had higher TBA values than salmon treated only with CO_2_. It has been established that HHP can induce lipid oxidation in salmon [20,41], and that such changes are usually observed at pressures >200 MPa. The use of HHP up to 150 MPa and modified atmospheres (50% CO_2_ + 50% O_2_) showed that the initial TBA value was affected neither by HPP nor the presence of CO_2_; after 14 days, the samples subjected to modified atmospheres and pressure exhibited higher oxidation levels than the control samples [40].

### 3.3. Effect of Carbon Dioxide (CO_2_) and High Hydrostatic Pressure (HHP) Treatment on Color and Texture of Coho Salmon

One of the major concerns associated with techniques to preserve salmon is the changes they might make to its appearance, especially the color, because it is one of the most important quality parameters. Considerable attention was directed to changes in the color of salmon produced by HHP, CO_2_, and combined treatments at the beginning and end of refrigerated storage. Table 3 shows the changes in lightness (L*), red/green (a*), and yellow/blue (b*) values on days 0 and 10. The L* value for an untreated fillet was 46.0 ± 0.0, which increased when using CO_2_ content >70%. The highest increase for this parameter was observed when the HHP treatment was applied without CO_2_ and with an L* value of 59.7 ± 0.3. It has been previously observed that the HHP treatment increases L*, which gives a “cooked” appearance to the fish when applying pressures >150 MPa; this result is commonly attributed to globin and myofibrillar denaturation [13,42]. The combined HHP and CO_2_ treatment was able to mitigate the increase in the L* value produced by pressure, as observed in treatments 150/50, 150/70, and 150/100. Treatments with 70% CO_2_ did not suffer any change in L* during refrigerated storage, regardless of the pressure applied; the L* value of the sample without pressure was closer to the fresh control sample.

The red color, indicated by the a* value, was 31.7 ± 0.1 in the fresh control sample. Most of the treatments caused small differences in color (< 3), except for the 0/100 treatment in which there was a high increase in red color. The reddish color in salmon is mainly attributed to the presence of carotenoids in the fish muscle (i.e., astaxanthin and cataxanthine) [43]; the oxidative status of hemeproteins has also had significant effects on the reddish color of salmon. Storage under anoxygenic conditions can prevent hemoglobin and myoglobin oxidation, which maintains the reddish color of salmon and even increases redness when stored in CO atmospheres [44]. Hemeprotein oxidation could also explain the significant decrease in the a* value corresponding to the untreated sample during refrigerated storage. After 10 days of storage, treated samples maintained a* values similar to fresh untreated salmon; the 0/50 treatment did not show any significant difference when compared with day 0. On the contrary, there was a significant decrease in a* in the untreated fish after 10 days.

The yellowness of salmon fillets (b*) was affected the most by the HHP treatment, decreasing from 33.8 ± 0.1 to 28.6 ± 1.7. The HHP and CO_2_ combination limited color loss produced by HPP on day 0. All treatments were able to maintain b* values similar to the fresh untreated control sample.

Total color change ∆E was evaluated as the difference between samples from each treatment and the fresh untreated salmon (control on day 0). All applied treatments caused very distinctive differences (∆E > 3) in the color of salmon fillets. The use of CO_2_ -free HHP produced the main color differences immediately after treatment. After 10 days of refrigerated storage, the untreated sample exhibited the highest ∆E (11.3 ± 0.3), which was mainly attributed to the decrease in redness and yellowness. This was followed by treatments 150/0 (9.9 ± 1.2) and 150/100 (9.6 ± 0.4), which were mainly produced by the high L* value of these samples. Salmon fillets treated with only CO_2_ retained more similarities in color to fresh salmon after storage.

As for the texture, hardness was not significantly affected by HHP or CO_2_ treatments compared with the control sample (p > 0.05). In this case, only three homogeneous groups were obtained, 0/70, 150/70, and 0/0-0/50-0/100-150/0-150/50-150/100. Hardness decreased during storage time, and a significant difference (p < 0.005) was observed among treatments after 10 days for all treatments, except 150/100 (*p* > 0.05). The 150/70 treatment obtained the lowest hardness value. Cheng et al. [4] mentioned that fish muscle texture was influenced by several factors, such as pH, rigor mortis, proteolysis, and temperature during storage. In the present study, the 150/100 and 150/0 treatments showed the lowest hardness reductions from days 0 to 10 (Table 4). The decrease in hardness could be attributed to the activation of collagenases during storage. Hultmann et al. [9] indicated that higher collagenase activity is related to lower hardness in Atlantic cod. We observed no collagenase activity on day 0; it increased during storage and could cause reduced hardness. Despite the reduced collagenase activity exhibited by all the treated salmon fillets compared with the untreated sample, no significant difference in hardness among treatments was observed on day 10. The only sample that did not show any significant difference in texture between days 0 and 10 was the 150/100 treatment, which also had the lowest collagenase activity at the end of storage. Delbarre-Ladrat et al. [45] suggested that the action of metalloproteases (collagenase and gelatinase, among others) could affect the significant loss of textural properties because these proteases would degrade muscle tissue proteins such as collagen and cytoskeletal proteins; therefore, the sarcolemma is connected to the extracellular matrix when the tissue exhibits a level of deterioration such as occurs during storage.

Both springiness and cohesiveness were significantly affected by treatments on days 0 and 10 (*p* < 0.05); values are markedly similar and it would not be appropriate to state that there is a clear trend. When considering samples between days 0 and 10, the ANOVA performed by comparing the springiness and cohesiveness values implied a significant influence of the treatments on these parameters (*p* < 0.05). In some cases, no significant difference was observed, such as in the 0/50 and 0/100 treatments for springiness and the 0/70 and 0/100 treatments for cohesiveness. Yagiz et al. [46] observed no significant changes for hardness, adhesiveness, gumminess, and chewiness in Atlantic salmon fillets after they were treated at 150 MPa for 15 minutes compared with untreated salmon; however, significant differences were found for springiness and cohesiveness. Briones-Labarca et al. [23] only reported a significant HHP effect on hardness and springiness of red abalone during refrigerated storage. Ayala et al. [47] observed changes in the textural properties of sea bream during refrigerated storage, which can be attributed to protein denaturation.

### 3.4. Effect of CO_2_ and HHP Treatment on Microbial Behavior and Shelf-Life Extension of Coho Salmon

Mean initial aerobic mesophilic (AMB) and aerobic psychrophilic bacteria (APB) in coho salmon were < 2 log CFU/g; for lactic acid bacteria and *Pseudomonas* spp., the initial values were below the detection limit for all conditions (Table 5). *Shewanella putrefaciens* and *Clostridium perfringens* were not detected in any sample at any time during refrigerated storage. Initial AMB and APB counts were in the range encountered in other studies involving salmon [26,43,48] and other marine species [49]. The low bacterial counts indicate excellent microbial quality; they are attributed to good hygiene practices from slaughter to the processing stage. Immediately after CO_2_ + HHP combined treatments, initial bacterial counts for AMB decreased compared with the untreated sample; this was only significant when applying > 70% CO_2_. Initial APB counts significantly decreased to non-detectable levels (*p* < 0.05) in all combined treatments.

Table 6 summarizes the effect of CO_2_ + HHP treatments on lag phase (λ), maximum specific growth rate (μmax), and shelf life (SL) of coho salmon fillets stored at 4 °C for 25 days. For all samples, the re-parameterized version of the Gompertz equation described microbial growth and determination coefficients (R^2^) ≥ 0.99. The SL of coho salmon fillets was affected by the CO_2_ content and not by the HPP treatment. According to the AMB and APB counts, the control samples showed SL between 9.48 and 9.72 days, which increased as the CO_2_ content increased to between 22.14 and 23.19 days with 100% CO_2_. The samples subjected to a combined treatment had similar behavior, with no significant differences between pressurized and non-pressurized samples at the same CO_2_ content level. Increased SL of samples was attributed to an increase in the lag phase (from 2 to 10 days) and a decrease in the μmax (from 0.63 to 0.35 1/day). Previous studies have shown that the HHP treatment between 135 and 200 MPa on salmon fillets decreased initial mesophilic bacterial counts, but this difference was not significant after 15 days of storage [50]. A 3 log cycle reduction in microbial counts was observed in Atlantic salmon fillets pressurized at 150 MPa; the longer pressurization time used in that study (15 minutes) could explain those results [47]. The HHP treatment <200 MPa is not usually effective in extending salmon SL [26]; in the present study, there is no synergistic effect when HHP is combined with CO_2_ to extend SL during storage. The HHP treatments on fish samples are commonly applied between 150 and 450 MPa and higher pressures are related to better microbial inactivation; they are also associated with significant changes in physicochemical and sensory properties [13,51]. The CO_2_ inhibits aerobic bacterial growth (*Pseudomonas* spp.), and it also dissolves in the product and decreases the pH value [48]. Different treatments involving changes in product atmosphere have been shown to significantly improve microbial quality and reduce AMB and APC in a variety of seafood species. These include treatments with ozone in shrimp, scald fish, musky octopus, and cod [49], modified atmospheres in salmon, hake, cuttlefish, chub mackerel, and yellow gurnard [48,52], and vacuum packaging in sardines, smoked salmon, and fresh eel [53]. Aponte et al. [49] noted that the sensitivity of microorganisms to gaseous bactericidal treatments (O_3_, CO_2_, gNO, N_2_, among others) varies according to the food matrix, the microbial species and strain, the method of applying and/or combined methods according to devices and procedures, and the methods adopted for the assessment of the antimicrobial efficacy.

For LAB and Pseudomonas, the lag phase also increased with increasing CO_2_ content, especially for *Pseudomonas* spp. (Figure 2). Pseudomonas are strict aerobic bacteria, so it is reasonable to find lower growth with a decrease in the oxygen content. During the 25 days of storage, Pseudomonas counts for fillets under modified atmospheres were <6 log CFU/g. The counts for LAB reached maximum growth on day 18 with 8.43 log in the control sample.

## 4. Conclusions

The main aim of this research study was to assess the effect of high hydrostatic pressure + carbon dioxide (HHP + CO_2_) on enzyme inactivation (proteases, lipases, and collagenases), physicochemical properties, and shelf life of coho salmon fillets. The added percentage of CO_2_ has a synergistic effect with HHP processing on endogenous enzyme inactivation. The deactivating effect of CO_2_ + HHP on enzyme activity was more significant in collagenase. Treatments with 100% CO_2_ limited the increase in TVB-N and the HHP treatments effectively limited the increase in TMA values. The TPA results showed that the HHP + CO_2_ treatments have no influence on hardness, but a slight influence on springiness and cohesiveness, for which lower mean values were found on day 0 for samples treated with HHP + CO_2_ compared with samples treated only with CO_2_. All treatments caused a significant change in the color of salmon (∆E >3), especially when only applying HHP. The CO_2_ treatments at 50%, 70%, and 100% without pressure were usually more effective at maintaining a color similar to fresh untreated salmon on days 0 and 10; this could be attributed to the protection against hemeprotein oxidation. Despite this situation, the combined treatment limited color change caused by HHP during storage. The microbial shelf life of salmon fillets increased with the use of CO_2_ and not with the HHP treatment; therefore, no synergy among treatments under the study conditions could be implied. As CO_2_ increased the lag phase of mesophilic and psychrophilic microorganisms, the maximum growth rate decreased. Further work on the kinetic modeling of endogenous enzyme inactivation (protease, lipase, and collagenase) is required to understand the effect of these technologies on enzymatic activity. In addition, the analysis of changes in secondary and tertiary structures of enzymes under study could explain the inactivation mechanisms of combined HHP + CO_2_ processing. Higher pressure levels or holding times could be used to improve inactivation, but these could negatively affect quality parameters such as color. We suggest that these combined technologies could also be useful in other processed salmon products, such as smoked salmon, which should be assessed in further work.

## Figures and Tables

**Figure 1 foods-09-00273-f001:**
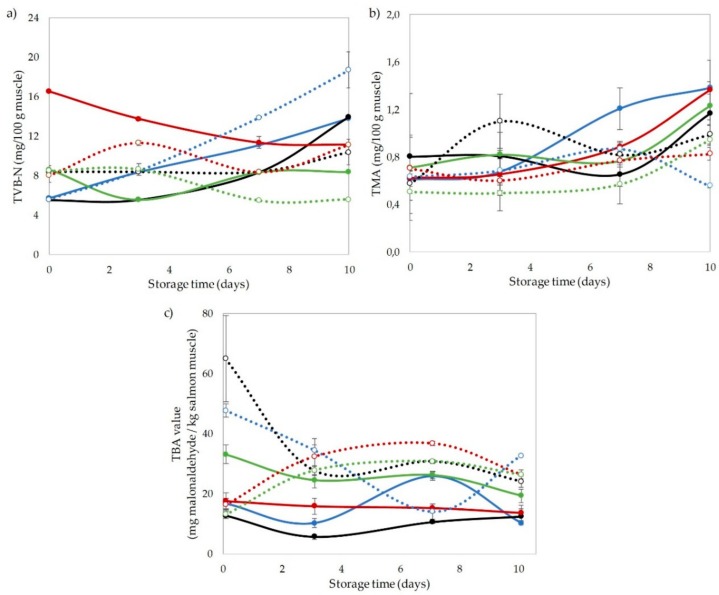
Effect of treatment and refrigerated storage time on (**a**) total volatile basic nitrogen (TVB-N), (**b**) trimethylamine (TMA), and (**c**) thiobarbituric acid (TBA) index for coho salmon fillets. Different lowercase letters indicate significant differences among treatments (*p* < 0.05). Different uppercase letters indicate significant (*p* < 0.05) differences among days of storage. (●) 0 MPa/0% CO_2_; (●) 0 MPa/50% CO_2_; (●) 0 MPa/70% CO_2_; (●) 0 MPa/100% CO_2_; (○) 150 MPa/0% CO_2_; (○) 150 MPa/50% CO_2_; (○) 150 MPa/70% CO_2_; (○) 150 MPa/100% CO_2_.

**Figure 2 foods-09-00273-f002:**
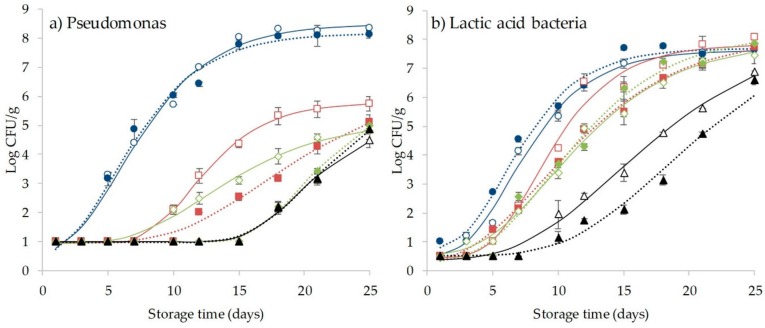
Growth curve of (**a**) Pseudomonas and (**b**) lactic acid bacteria (LAB) in treated and untreated coho salmon fillets. Symbols and lines (solid—without high hydrostatic pressure (HHP); dashed—with HHP) represent observed and modeled (re-parameterized version of Gompertz equation) values, respectively. (○) 0 MPa/0% CO_2_; (□) 0 MPa/50% CO_2_; (◊) 0 MPa/70% CO_2_; (∆) 0 MPa/100% CO_2_; (●) 150 MPa/0% CO_2_; (■)150 MPa/50% CO_2_; (♦) 150 MPa/70% CO_2_; (▲) 150 MPa/100% CO_2_.

**Table 1 foods-09-00273-t001:** Effect of treatment and refrigerated storage time on enzyme activity of coho salmon fillets.

	Treatment (MPa/CO_2_)	Storage Time (Days)			
		0	3	7	10
	0/0	0.397 ± 0.002 ^fA^	0.241 ± 0.001 ^aB^	0.231 ± 0.011 ^abB^	0.364 ± 0.006 ^bC^
Protease activity (IU)	0/50	0.280 ± 0.004 ^cA^	0.124 ± 0.004 ^bB^	0.439 ± 0.006 ^cC^	0.217 ± 0.004 ^aD^
	0/70	0.194 ± 0.007 ^bA^	0.414 ± 0.019 ^cB^	0.281 ± 0.015 ^dC^	0.369 ± 0.030 ^bB^
	0/100	0.341 ± 0.003 ^eA^	0.179 ± 0.003 ^dB^	0.529 ± 0.035 ^eC^	0.200 ± 0.002 ^aB^
	150/0	0.389 ± 0.014 ^fA^	0.212 ± 0.001 ^eB^	0.354 ± 0.002 ^fC^	0.389 ± 0.011 ^bA^
	150/50	0.317 ± 0.008 ^dA^	0.111 ± 0.004 ^bB^	0.248 ± 0.008 ^adA^	0.756 ± 0.103 ^dC^
	150/70	0.389 ± 0.006 ^fA^	0.121 ± 0.004 ^bB^	0.202 ± 0.003 ^bgC^	0.341 ± 0.001 ^bD^
	150/100	0.163 ± 0.004 ^aA^	0.258 ± 0.013 ^aB^	0.168 ± 0.005 ^gA^	0.643 ± 0.013 ^cD^
Lipase activity (IU)	0/0	0.379 ± 0.007 ^aA^	0.428 ± 0.006 ^aB^	0.296 ± 0.002 ^deC^	0.314 ± 0.013 ^dC^
	0/50	0.279 ± 0.003 ^bA^	0.382 ± 0.002 ^aB^	0.301 ± 0.012 ^cdA^	0.270 ± 0.028 ^cA^
	0/70	0.246 ± 0.005 ^cA^	0.280 ± 0.002 ^bcB^	0.280 ± 0.002 ^eB^	0.271 ± 0.015 ^cB^
	0/100	0.390 ± 0.001 ^dA^	0.419 ± 0.058 ^aA^	0.219 ± 0.002 ^fB^	0.186 ± 0.003 ^aB^
	150/0	0.290 ± 0.006 ^eA^	0.290 ± 0.005 ^bA^	0.370 ± 0.007 ^aB^	0.207 ± 0.006 ^aC^
	150/50	0.607 ± 0.005 ^fA^	0.237 ± 0.227 ^bcB^	0.289 ± 0.001 ^deC^	0.233 ± 0.003 ^bB^
	150/70	0.555 ± 0.005 ^gA^	0.227 ± 0.014 ^cB^	0.315 ± 0.006 ^cC^	0.250 ± 0.019 ^bcB^
	150/100	0.408 ± 0.008 ^hA^	0.399 ± 0.014 ^aA^	0.335 ± 0.006 ^bB^	0.402 ± 0.010 ^eA^
Collagenase activity (IU)	0/0	0.000 ^A^	1.675 ± 0.066 ^aB^	1.774 ± 0.068 ^aB^	2.458 ± 0.068 ^eD^
	0/50	0.000 ^A^	0.531 ± 0.015 ^deB^	0.843 ± 0.074 ^bC^	0.781 ± 0.053 ^dC^
	0/70	0.000 ^A^	0.543 ± 0.032 ^cdeB^	1.966 ± 0.004 ^cC^	0.421 ± 0.006 ^bD^
	0/100	0.000 ^A^	1.198 ± 0.022 ^bB^	0.200 ± 0.019 ^fC^	0.229 ± 0.021 ^aC^
	150/0	0.000 ^A^	0.691 ± 0.091 ^cB^	0.476 ± 0.016 ^deC^	0.266 ± 0.019 ^aD^
	150/50	0.000 ^A^	0.307 ± 0.023 ^fB^	0.551 ± 0.042 ^dC^	0.411 ± 0.022 ^bcBC^
	150/70	0.000 ^A^	0.464 ± 0.002 ^eBC^	0.396 ± 0.039 ^eB^	0.543 ± 0.061 ^cC^
	150/100	0.000 ^A^	0.664 ± 0.016 ^cdB^	0.634 ± 0.009 ^dB^	0.210 ± 0.023 ^aC^

Different lowercase letters indicate significant differences among treatments (*p* < 0.05). Different uppercase letters indicate significant differences among days of storage (*p* < 0.05).

**Table 2 foods-09-00273-t002:** Effect of treatment and refrigerated storage time on pH of coho salmon fillets.

Treatment (MPa/CO_2_)	Storage Time (Days)			
	0	3	7	10
0/0	6.05 ± 0.10 ^aA^	6.22 ± 0.02 ^aB^	6.19 ± 0.02 ^aAB^	6.20 ± 0.01 ^abB^
0/50	6.16 ± 0.02 ^abA^	6.16 ± 0.03 ^bA^	6.15 ± 0.01 ^bA^	6.08 ± 0.15 ^bcA^
0/70	6.16 ± 0.02 ^abAB^	6.20 ± 0.03 ^abA^	6.07 ± 0.02 ^cB^	5.90 ± 0.09 ^dC^
0/100	6.13 ± 0.03 ^abA^	6.20 ± 0.01 ^abB^	6.14 ± 0.01 ^bA^	6.00 ± 0.02 ^cdC^
150/0	6.35 ± 0.03 ^bA^	6.21 ± 0.01 ^abB^	6.24 ± 0.01 ^dB^	6.11 ± 0.01 ^abcC^
150/50	6.15 ± 0.02 ^abA^	6.21 ± 0.01 ^abB^	6.20 ± 0.01 ^aB^	6.28 ± 0.02 ^aC^
150/70	6.30 ± 0.01 ^bcA^	6.21 ± 0.03 ^abB^	6.22 ± 0.02 ^adB^	6.24 ± 0.01 ^abB^
150/100	6.41 ± 0.13 ^cA^	6.22 ± 0.01 ^aB^	6.28 ± 0.02 ^eAB^	6.17 ± 0.02 ^abB^

Different lowercase letters indicate significant differences among treatments (*p* < 0.05). Different uppercase letters indicate significant differences among days of storage (*p* < 0.05).

**Table 3 foods-09-00273-t003:** Effect of treatment and refrigerated storage time on color parameters of coho salmon fillets.

Treatment (MPa/CO_2_)	Color Parameters							
	L*		a*		b*		∆E	
	Day 0	Day 10	Day 0	Day 10	Day 0	Day 10	Day 0	Day 10
0/0	46.0 ± 0.0 ^aA^	49.6 ± 0.3 ^aB^	31.7 ± 0.1 ^aA^	23.3 ± 0.5 ^aB^	33.8 ± 0.1 ^aA^	27.2 ± 0.1 ^aB^	0.0 ^aA^	11.3 ± 0.3 ^aB^
0/50	46.1 ± 0.2 ^aA^	48.5 ± 0.1 ^Bb^	33.4 ± 0.2 ^abA^	33.4 ± 0.4 ^bcA^	36.2 ± 0.6 ^bA^	32.2 ± 0.6 ^bcB^	3.8 ± 0.3 ^bcA^	3.4 ± 0.3 ^dA^
0/70	48.3 ± 0.3 ^bA^	48.0 ± 0.0 ^cA^	33.8 ± 0.6 ^bA^	32.3 ± 0.1 ^cB^	34.2 ± 0.6 ^abA^	31.9 ± 0.1 ^bB^	3.3 ± 0.2 ^bA^	3.2 ± 0.3 ^dA^
0/100	50.6 ± 0.0 ^cA^	48.5 ± 0.2 ^bB^	38.1 ± 0.2 ^cA^	33.2 ± 0.2 ^bcdB^	33.0 ± 0.1 ^aA^	34.0 ± 2.3 ^bcdA^	4.7 ± 0.0 ^cdA^	3.4 ± 0.4 ^dB^
150/0	59.7 ± 0.3 ^dA^	53.1 ± 0.1 ^dB^	28.3 ± 1.7 ^dA^	36.8 ± 0.1 ^eB^	28.6 ± 1.7 ^cA^	34.6 ± 0.4 ^cdB^	15.2 ± 0.7 ^fA^	9.9 ± 1.2 ^abB^
150/50	50.1 ± 0.1 ^cA^	51.5 ± 0.2 ^cB^	32.3 ± 0.2 ^abA^	33.9 ± 0.1 ^bdB^	32.1 ± 0.1 ^adA^	33.9 ± 0.4 ^bcdB^	4.4 ± 0.1 ^cdA^	5.8 ± 0.2 ^cB^
150/70	52.7 ± 0.1 ^eA^	52.3 ± 0.1 ^eA^	31.7 ± 0.5 ^aA^	29.4 ± 0.4 ^fB^	30.5 ± 0.7 ^cdA^	35.0 ± 0.6 ^dB^	7.5 ± 0.6 ^eA^	6.7 ± 0.2 ^cA^
150/100	49.2 ± 0.1 ^fA^	55.2 ± 0.2 ^fB^	28.8 ± 0.4 ^dA^	32.5 ± 0.6 ^cdB^	36.3 ± 0.5 ^bA^	32.7 ± 0.3 ^bcdB^	5.4 ± 0.2 ^dA^	9.6 ± 0.4 ^bB^

L*: lightness; a*: red/green; b*: yellow/blue; ∆E: total color difference. Different lowercase letters indicate significant differences among treatments (*p* < 0.05). Different uppercase letters indicate significant differences among days of storage (*p* < 0.05).

**Table 4 foods-09-00273-t004:** Effect of treatment and refrigerated storage time on texture parameters of coho salmon fillets.

Treatment (MPa/CO_2_)	Texture Parameters					
	Hardness (N)		Springiness (cm)		Cohesiveness	
	Day 0	Day 10	Day 0	Day 10	Day 0	Day 10
0/0	12.92 ± 2.47 ^abA^	7.57 ± 1.03 ^abB^	0.57 ± 0.07 ^abcA^	0.67 ± 0.05 ^aB^	0.41 ± 0.02 ^aA^	0.47 ± 0.04 ^abB^
0/50	11.53 ± 3.22 ^abA^	6.90 ± 1.57 ^abB^	0.64 ± 0.04 ^aA^	0.65 ± 0.04 ^aA^	0.52 ± 0.04 ^bA^	0.42 ± 0.07 ^aB^
0/70	13.45 ± 1.78 ^aA^	8.22 ± 1.86 ^abB^	0.62 ± 0.00 ^abA^	0.67 ± 0.03 ^aB^	0.44 ± 0.04 ^aA^	0.50 ± 0.08 ^abA^
0/100	14.39 ± 3.51 ^abA^	7.22 ± 0.99 ^abB^	0.58 ± 0.01 ^bcA^	0.61 ± 0.04 ^aA^	0.42 ± 0.09 ^aA^	0.48 ± 0.10 ^abA^
150/0	11.39 ± 1.32 ^abA^	8.04 ± 1.19 ^abB^	0.51 ± 0.03 ^cA^	0.59 ± 0.08 ^aB^	0.40 ± 0.04 ^aA^	0.50 ± 0.07 ^abB^
150/50	10.18 ± 3.21 ^abA^	6.63 ± 1.85 ^bB^	0.50 ± 0.02 ^cA^	0.58 ± 0.05 ^aB^	0.51 ± 0.06 ^aA^	0.55 ± 0.10 ^bB^
150/70	14.84 ± 1.18 ^bA^	5.36 ± 1.10 ^bB^	0.50 ± 0.05 ^cA^	0.60 ± 0.04 ^aB^	0.35 ± 0.09 ^aA^	0.47 ± 0.04 ^abB^
150/100	14.07 ± 3.45 ^abA^	10.54 ± 4.80 ^aA^	0.46 ± 0.05 ^dA^	0.60 ± 0.02 ^aB^	0.25 ± 0.05 ^cA^	0.49 ± 0.04 ^abB^

Different lowercase letters indicate significant differences among treatments (*p* < 0.05). Different uppercase letters indicate significant differences between days of storage (*p* < 0.05).

**Table 5 foods-09-00273-t005:** Effect of treatment on initial counts of studied microbial populations (log CFU/g) in coho salmon fillets.

Treatment (MPa/CO_2_)	Microbial Groups			
	Aerobic Mesophilic Microorganisms	Aerobic Psychrophilic Microorganisms	Lactic Acid Bacteria	*Pseudomonas* spp.
0/0	1.69 ± 0.09 ^ab^	1.67 ± 0.06 ^a^	< 1.00 ^a^	< 2.00 ^a^
0/50	1.46 ± 0.15 ^ac^	1.42 ± 0.10 ^a^	< 1.00 ^a^	< 2.00 ^a^
0/70	1.56 ± 0.07 ^abc^	1.42 ± 0.10 ^a^	< 1.00 ^a^	< 2.00 ^a^
0/100	1.75 ± 0.05 ^ab^	1.46 ± 0.15 ^a^	< 1.00 ^a^	< 2.00 ^a^
150/0	1.76 ± 0.06 ^b^	1.65 ± 0.16 ^a^	1.00 ± 0.00 ^a^	< 2.00 ^a^
150/50	1.46 ± 0.15 ^ac^	< 1.00 ^b^	< 1.00 ^a^	< 2.00 ^a^
150/70	1.26 ± 0.24 ^c^	< 1.00 ^b^	< 1.00 ^a^	< 2.00 ^a^
150/100	1.36 ± 0.10 ^c^	< 1.00 ^b^	< 1.00 ^a^	< 2.00 ^a^

Different letters indicate significant differences among treatments (*p* < 0.05).

**Table 6 foods-09-00273-t006:** Effect of treatment on initial counts of studied microbial populations (log CFU/g) in coho salmon fillets.

Treatment (MPa/CO_2_)	Kinetic Parameters					
	Aerobic Mesophilic			Aerobic Psychrophilic		
	λ (Days)	μmax (1/Days)	SL (Days)	λ (Days)	μmax (1/Days)	SL (Days)
0/0	2.33 ± 1.25 ^a^	0.63 ± 0.07 ^a^	9.72 ± 1.04 ^a^	2.30 ± 1.48 ^ab^	0.68 ± 0.10 ^a^	9.48 ± 1.35 ^a^
0/50	5.10 ± 1.32 ^bc^	0.60 ± 0.09 ^ab^	13.66 ± 1.51 ^b^	4.79 ± 0.69 ^ab^	0.54 ± 0.04 ^abc^	14.05 ± 0.78 ^b^
0/70	5.97 ± 1.38 ^c^	0.48 ± 0.07 ^abc^	15.87 ± 1.61 ^b^	4.72 ± 1.84 ^ab^	0.42 ± 0.05 ^cd^	16.36 ± 1.51 ^b^
0/100	10.46 ± 0.66 ^d^	0.35 ± 0.02 ^c^	23.19 ± 0.93 ^c^	10.31 ± 1.15 ^c^	0.37 ± 0.05 ^cd^	22.14 ± 1.63 ^c^
150/0	2.47 ± 1.32 ^ab^	0.63 ± 0.09 ^a^	9.73 ± 1.22 ^a^	1.95 ± 1.77 ^a^	0.62 ± 0.09 ^ab^	9.63 ± 1.35 ^a^
150/50	4.99 ± 1.43 ^abc^	0.46 ± 0.06 ^bc^	16.01 ± 1.69 ^b^	3.02 ± 0.98 ^ab^	0.51 ± 0.03 ^bcd^	15.29 ± 0.89 ^b^
150/70	4.90 ± 1.02 ^abc^	0.44 ± 0.04 ^c^	16.71 ± 1.05 ^b^	5.21 ± 1.20 ^b^	0.55 ± 0.05 ^abc^	16.26 ± 1.53 ^b^
150/100	9.72 ± 1.04 ^d^	0.43 ± 0.06 ^c^	22.72 ± 2.04 ^c^	9.11 ± 1.15 ^c^	0.43 ± 0.05 ^cd^	22.40 ± 1.98 ^c^

λ (Days): lag phase; μmax (1/Days): maximum specific growth rate; SL (Days): shelf life. Different letters indicate significant differences among treatments (*p* < 0.05).

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
