# Peer review of "Combined Treatments of High Hydrostatic Pressure and CO2 in Coho Salmon (Oncorhynchus kisutch): Effects on Enzyme Inactivation, Physicochemical Properties, and Microbial Shelf Life"

_foods, 2020, doi:10.3390/foods9030273_

Round 1

Reviewer 1 Report

The authors describes combined effect of HHP and CO2 on collagenase, protease, and lipase enzymatic activity, physico chemical properties, and microbial shelf life of coho salmon (Oncorhynchus kisutch). The results are interesting because deal with an important topic for seafood industry. There are several researchs with contradictory results, including data regarding HHP and CO2 combined processing to increase the shelf life of marine products while inactivating microbial growth and endogenous enzymedeterioration. I think that the manuscript can be improved with some items indicated below. Some papers could be cited to improve the discussion. I think it is interesting to discuss data with results of:

Impact of gaseous ozone coupled to passive refrigeration system to maximize shelf-life and quality of four different fresh fish products. LWT Volume 93, July 2018, Pages 412- 419. https://doi.org/10.1016/j.lwt.2018.03.073

and

Effect of high pressure processing and cooking treatment on the quality of Atlantic salmon Food Chemistry Volume 116, Issue 415 October 2009 Pages 828-835. https://doi.org/10.1016/j.foodchem.2009.03.029

The authors also must indicate why only the presence of Clostridium perfringens was tested at the end of the storage period. Clostridium botulinum type E? Isn't this also a risk for adding CO2?

Author Response

  1. We take the reviewer recommendations and both papers have been included in the manuscript to improve the discussion.
  2. The presence of Clostridium perfringens was only investigated at the end of the storage period because the storage temperature used in this investigation was 4°C. As is well known, Clostridium perfringens does not grow at temperatures <10°C (Huang and Li, 2020). However, considering that a CO2-rich atmosphere can stimulate the growth of Clostridium sp. it was still considered to check for its presence, but only at the end of the incubation period. It should also be noted that in preliminary studies, fresh salmon samples always showed perfringens counts <10 CFU/g.

    Lihan Huang, Changcheng Li (2020) Growth of Clostridium perfringens in cooked chicken during cooling: One-step dynamic inverse analysis, sensitivity analysis, and Markov Chain Monte Carlo simulation. Food Microbiology 85: 103285. https://doi.org/10.1016/j.fm.2019.103285.

Changes have been highlighted in blue in the manuscript. Please see the attachment.

Reviewer 2 Report

Dear authors,

The description of the TVB-N, TMA and TBA method is short and only refers to the publication citation. Please complete the methodology:

Were distillation or extraction methods used for the TBA test according to Vyncke (1970)? Was an antioxidant additive used for testing? At what wavelength the absorbance of reaction products was measured.

In the official method (AOCS 971.14) TMA is designated as the picrate salt. What was the concentration of TCA and what was the ratio of muscle dilution.

What method used for TVB-N determination:with PCA as in Aubourg et al. 2010 or with MgO?

Author Response

We take the reviewer recommendations and the methodologies for TVB-N, TMA and TBARS have been completed.

  1. For TBARS determination an extraction method was used according to the methodology of Vyncke (1970) modified by Sørensen, G.; Storgaard Jørgensen (1996). Propylgallate at 0.1% was used as an antioxidant and the absorbance was measured at 532 nm.
  2. To determine TMA the concentration of TCA used was 7.5%. The ratio of muscle dilution was 1:2 (6.5 g salmon, 12.5 g TCA)
  3. The method for TVB-N determination was steam distillation with MgO according to the methodology proposed by Gallardo et al., 2010.

Changes have been highlighted in blue in the manuscript. Please see the attachment.
